# Integrated Probe System for Measuring Soil Carbon Dioxide Concentrations

**DOI:** 10.3390/s23052580

**Published:** 2023-02-26

**Authors:** Sammy Hassan, Ryan M. Mushinski, Tilahun Amede, Gary D. Bending, James A. Covington

**Affiliations:** 1School of Engineering, University of Warwick, Coventry CV4 7AL, UK; 2School of Life Sciences, University of Warwick, Coventry CV4 7AL, UK; 3International Crops Research Institute for the Semi-Arid Tropics (ICRISAT), Addis Ababa P.O. Box 5689, Ethiopia

**Keywords:** soil measurements, Wireless Sensor Networks (WSNs), environmental monitoring, internet-of-things (IoT)

## Abstract

This article outlines the design and implementation of an internet-of-things (IoT) platform for the monitoring of soil carbon dioxide (CO_2_) concentrations. As atmospheric CO_2_ continues to rise, accurate accounting of major carbon sources, such as soil, is essential to inform land management and government policy. Thus, a batch of IoT-connected CO_2_ sensor probes were developed for soil measurement. These sensors were designed to capture spatial distribution of CO_2_ concentrations across a site and communicate to a central gateway using LoRa. CO_2_ concentration and other environmental parameters, including temperature, humidity and volatile organic compound concentration, were logged locally and communicated to the user through a mobile (GSM) connection to a hosted website. Following three field deployments in summer and autumn, we observed clear depth and diurnal variation of soil CO_2_ concentration within woodland systems. We determined that the unit had the capacity to log data continuously for a maximum of 14 days. These low-cost systems have great potential for better accounting of soil CO_2_ sources over temporal and spatial gradients and possibly flux estimations. Future testing will focus on divergent landscapes and soil conditions.

## 1. Introduction

Anthropogenic activities, including intensive agriculture and land-use changes, have contributed to the exponential rise in atmospheric carbon dioxide (CO_2_) [1]. Evidence suggests that these two factors may account for 25% of all CO_2_ emissions since 2007 [2]. Gross soil respiration, which is the positive flux of CO_2_ from the soil to the atmosphere, is estimated at 60 Gt CO_2_-C yr^−1^—representing one of the largest carbon sources. Between 60–90% of this CO_2_ is sourced from microbial activity in soil [3]. Practices that lead to soil disturbance, such as tillage for crop production and deforestation, can stimulate microbial activity, leading to pulses of CO_2_ flux. Current methods for measuring soil respiration commonly utilise headspace sampling of small chambers placed on the soil surface. Air within the headspace is sampled over a set period of time, analysed using a gas chromatograph or infrared gas analyser, and then the concentration is regressed over sampling time to calculate flux [4]. More modern systems utilise headspace recirculation to automatically calculate CO_2_ flux [5]. While well-represented in the literature, these methods are time-intensive and generally lack spatial resolution. Furthermore, the analytical components can be cost-prohibitive. Thus, there is a pressing need to develop systems that are cost-effective, can provide spatial and temporal resolution, can be utilized for flux estimation, and have data acquisition programs that are easily accessible.

The increased development and availability of the internet has spawned the technology of connected devices and the Internet of Things (IOT). IOT describes the integration and communication between intelligent objects, commonly referred to as “Things” [6]. Similarly, the advancement of embedded computing, downsizing of electronic devices and the development of IOT has contributed to the technology of Wireless Sensor Networks (WSNs) [7,8,9,10]. WSNs form a large component of IOT systems and have found applications in environmental monitoring, healthcare, safety applications, air quality applications, gas monitoring and agriculture [6,8,10,11,12,13,14,15]. WSNs usually consist of a series of static or mobile nodes that transmit data to single or multiple base stations, which themselves can be static or mobile. There is also no specific definition of the technology that can be used to transmit data to a base station. However, common techniques include Wi-Fi, GSM, Bluetooth and LoRa [16,17,18,19,20,21,22,23].

With the increasing impact of WSNs, it is not surprising that such systems have been used in applications linked with agriculture, including soil analysis. To this end, a number of research groups have reported the development of different WSNs specifically targeted towards precision agriculture [24,25,26,27,28]. Even within the agricultural sector, the range of different potential applications of WSNs is considerable. These range from smart fertilisation systems [24,29] to pest control [30,31,32] and early detection of disease [32,33]. For the latter two systems, the most common approach is to use some form of optical system to take readings of the crop, which can be used to inform the farmer/grower of any issues [25,34]. This is sometimes combined with UAVs that act as both a base station and as a means of capturing additional information on the crop [35,36,37]. More closely related to the work presented here is linkage to soil irrigation. Researchers have investigated the use of soil probes for smart irrigation systems using soil moisture to control the watering system [21,22,38]. This is similar to the majority of WSN networks for soil, where most researchers measure soil parameters including temperature, humidity, pH and moisture to monitor crop health [18,21,22,27]. For example, a recent paper described a system deploying soil measurement probes detecting temperature, humidity and pH in a WSN network, communicating to a central hub via Wi-Fi. The data subsequently were transmitted through the internet to a cloud server and provided results from different agricultural sites [22].

The measurement of CO_2_ using WSN has almost solely been for air quality monitoring [13,15,23,39], with some agriculture-based applications, including smart cattle sheds [40]. However, work on the direct measurement of soil-specific CO_2_ with a WSN is much more limited. Most research in this sector lacks direct soil CO_2_ measurements, using only air CO_2_ measurements along with ancillary soil sensors, such as temperature and humidity [16,21,22,25]. For in situ soil CO_2_ concentration measurements, there are two main commercial systems. These are produced by Vaisala (GMP343) [41] and Eosense (eosGPCO_2_) [42] and offer a means to monitor in situ CO_2_ levels at different soil depths. These units are relatively expensive (including supplementary costs associated with separate waterproofing components), require a continuous power supply and need to be wired to a base station (or other communication module), making them less user-friendly for a range of soil-monitoring applications. Furthermore, these commercial units offer no information regarding the inner working of their devices and the CO_2_ sensors used. Therefore, the aim of this project was to create a battery-powered/portable monitoring system for measuring soil CO_2_ concentration and related environmental parameters to address the spatial coverage, connectivity and cost limitations of current systems. In this paper, we describe the development of, to our knowledge, the first reported wireless and battery-powered IOT WSN unit that can measure CO_2_ concentration and other directly related environmental parameters of soil. We then present the results collected from three different field measurement scenarios to show the functionality of the unit.

## 2. Materials and Methods

Our main design requirement was for the probes to operate under environmental conditions of a wide range of terrestrial landscapes, providing spatial coverage across any given site. In use, the probes would be placed into the soil at different depths, vertically up to a maximum of 15 cm, or horizontally into a soil profile. Having sensors at various depths within a single location would facilitate post hoc flux calculation of CO_2_ transport through the soil profile using the flux gradient method [43]. Each probe would measure CO_2_ concentrations at the expected levels found in soil [44] for an extended duration running on batteries. The probes would also be required to measure environmental factors (temperature and humidity), which are key factors that regulate soil microbial activity and associated respiration. Soil–atmosphere carbon flux can also occur via loss of volatile organic compounds (VOC) originating from litter, plants and soil microbes, and furthermore VOCs may provide a convenient means of measuring soil microbial activity [45]. Therefore, the capacity to include measurement of soil VOC concentrations was also considered. Based on this requirement, we developed a system based on an array of sensor probes and nodes to collect local soil CO_2_ concentrations, which can then be transmitted to a single gateway. The gateway then transmits the data to a server/database that can be accessed at any time by users. The basic concept of our system is shown in Figure 1, with the sub-systems described in the following sections.

### 2.1. Sensor Probe

The sensor probe is the main measurement component of the monitoring system. It comprises of sensors to measure CO_2_ and environmental parameters, a microcontroller, a real-time clock, SD card back-up and communication module. A block diagram overview of the system is shown in Figure 2 with data flow arrows.

For CO_2_ monitoring, the most common approach is to use a sensor based on non-dispersive infra-red (NDIR) technology. There are many different commercial NDIR CO_2_ sensors currently available on the market. The sensors described here use an LED illumination, rather than an incandescent source, making the power consumption significantly lower. The LED CO_2_ sensors were sourced from GSS Ltd. (Cumbernauld, UK) due to their very low power consumption (< 4 mW) compared with typical NDIR CO_2_ sensors (>10 mW). It was also considered important to measure environmental factors, and so a combined temperature and humidity sensor was added, along with a total VOC sensor, sourced from Sensirion (Zurich, Switzerland). Communication with the main control system was performed using I^2^C and UART. Details of the sensors and main design components are provided in Table 1.

The main control board used a Microchip SAMD21G ARM M0+ microcontroller. This was chosen due to its reliability, common use and available software libraries. Communication between the probe and the main gateway was undertaken using LoRa. This was chosen over methods such as Bluetooth, as the distances between probes could be considerable (100 s of m) and the data rates are low, and thus do not require high-speed transfer. The units were fitted with external aerials to help communication across a field. As stated earlier, the unit was also fitted with a real-time clock, with separate back-up battery, to ensure that the timing of the probes was the same across the field and would not be lost in case of battery discharge. The main control system also contained a SD card for storing data in case of issues surrounding communication with the gateway system.

The main probe unit was divided into three printed circuit boards (PCBs), with one for the sensors, one for the main control electronics and one for communication and connections. The communication PCB contained the aerial connection, USB connector and a tri-colour LED to indicate the status of the unit. This solution ensured that the PCBs would fit into the selected housing for this project. The housing was a three-piece plastic unit, which was re-purposed from an Alphasense (Essex, UK) Electronic Diffusion Tube unit. The housing also contained a battery holder for 2 off 3.7 V 14,500 rechargeable batteries. A photograph of the casing and of the internal configuration are shown in Figure 3 and Figure 4. A 3D-printed protective cover was made for both ends of the probe to reduce water ingress while also allowing air movement into and out of the unit. The probe was 230 mm long and a maximum of 35 mm in diameter, and with batteries weighed 172 g.

The temperature, humidity and VOC sensor was used as received and calibrated by the manufacturer. The CO_2_ sensor also used the manufacturer’s calibration. However, a N_2_ re-calibration of the baseline was undertaken before the start of the experiments to ensure that all the sensors gave the same value (within measurement error provided by the manufacturer). Furthermore, the probes were checked against a reference CO_2_ monitor used to monitor our lab environment, and the readings were also within the manufacturer’s tolerance. Finally, the probes were programmed so that the microcontroller entered a low-power mode in between each measurement to reduce power consumption. Due to their lower power consumption, we were able to keep all the sensors fully powered when the unit was in operation to ensure the continual stability of the system.

### 2.2. Gateway

The purpose of the gateway is to connect to the sensor probes and receive data. Each probe was given a unique address, and therefore the gateway can communicate independently with each probe. In our solution, a Raspberry Pi 4 (Cambridge, UK) was used as a gateway, fitted with an Adafruit LoRa bonnet (model RFM96 @ 433 MHz, for Europe). Additionally, a mobile 4G cellular module was used that allowed the Raspberry Pi to upload data to a server without relying on a Wi-Fi or wired connection (Raspberry Pi hat). The setup required two programming scripts: one to manage the LoRa module (written in Python) and one to manage the cellular connection (Bash shell script), which were both executed at start-up. This setup is shown in Figure 5 below.

To protect the unit from the elements, the Raspberry Pi and associated hardware were fitted into a metal case (as shown in Figure 5, dimensions 55 × 105 × 165 mm and weight 700 g). This was fitted with an OLED screen to provide a visual indication of the status of the sensor probes and of the communication link.

### 2.3. Data storage and Dashboard

Here, we used a data storage platform that allowed remote input/output of data and could be seated on a server. Hosting was undertaken with Fasthosts (UK), which provided a simple means to set up a database, host background services such as APIs and host websites that can be used to view and access data. On the server, a MySQL database was used. Additionally, we used a Python framework (Flask) to build the API and manage the data being entered into the database. The benefit of using a remote database is that large amounts of data can be stored, sorted and made available to multiple users.

Table 2 summarises the programs we have implemented, where they were implemented, if they were managed by other software and how we accessed them for modification/development.

The dashboard pulls data directly from the database, in real time, so any issues can be flagged rapidly. The dashboard can be configured to display any part of the recorded data from any time. Figure 6 shows the time series alongside the latest recorded value (shown here rewound to the experiment dates). The website was hosted on a Virtual Private Server (VPS) provided by the UK company Fasthosts. The dashboard visuals were created using the Grafana web application running on the server.

### 2.4. Experimental Method

The probes were loaded with a program that takes a reading from every sensor, saves the data to an SD card and sends the data over LoRa radio when triggered. The trigger was set to go off once every minute; this was increased to two minutes after the initial deployment to prolong battery life. Two minutes was chosen as a compromise between saving power and maintaining high sample frequency. To protect the device from excess water ingress from above and the surrounding soil, a top cap was fitted with an umbrella-style cover, as shown in Figure 7. In total, eight probes were used in three deployments. No additional reference CO_2_ soil monitor was used within the experiments.

#### 2.4.1. Deployment 1

This deployment was undertaken over October–November 2021. For this deployment the probes were inserted into woodland soil (coordinates: 52.376134, −1.557528) within the University of Warwick grounds and were spread within a 10 m^2^ area (Figure 8). Soil here was freely draining, slightly acid loam, with texture ranging from clay loam to sandy loam. Overstory vegetation was dominated by secondary growth *Quercus robur* with a patchy understory of *Pteridium* spp. In this case, 4 probes were used for 5 days as part of an initial/first trial.

#### 2.4.2. Deployment 2

On the 13 July 2022, eight probes were deployed in a woodland at the Wellesbourne campus of the University of Warwick (coordinates: 52.211069, −1.610663). Soil here was the same composition as deployment 1, a freely draining, slightly acid loam, with texture ranging from clay loam to sandy loam. Overstory vegetation was an equal mix of *Quercus robur* and *Acer pseudoplatanus* was the understory dominated by *Hedera* spp. Two devices were placed at 5 cm and 10 cm every 5 m along a 20 m transect. After 6 days, the device’s driven depth was swapped (5 cm became 10 cm and vice versa). Additionally, an Onset HOBO data logger was placed adjacent to each pair and was placed 5 cm below the surface. The HOBO units are available off the shelf and offer commercial-grade data logging and wireless connectivity. These devices serve as a suitable comparison for our probes.

#### 2.4.3. Deployment 3

Nine probes were deployed on the 26 August at 4 pm at the same location as deployment 2; three groups of three devices were placed at 3, 10 and 15 cm depths. One device failed to take any readings (position 2, 5 cm) and another device failed after 3 days (position 3, 15 cm). The latter device was suspected to have suffered excess water ingress from flooding.

## 3. Results

### 3.1. Deployment 1

To show the ability of the probes to monitor CO_2,_ temperature, humidity and VOC index over the five-day experimental period, the dynamic responses of two pairs of sensors (one at 5 cm and the other at 10 cm) are shown in Figure 9, Figure 10, Figure 11 and Figure 12. Figure 9 shows the CO_2_ response from a pair of probes in open woodland (A and B) and under a tree (C and D), respectively.

All sensors showed initial CO_2_ readings that were close to ambient above-ground levels and then rapidly increased over a short period of time. There was a universal increase in concentration through the period of the experiment, which is likely related to the soil restabilizing from the disturbance associated with fitting the probes. Constructing a hole is likely to disturb soil and increase aeration, stimulating microbial activity and production of CO_2_ [46]. We expected the physical removal of a soil core and replacement with our sensor to stimulate CO_2_ emissions due to enhanced microbial activity at the soil–sensor interface. However, CO_2_ concentrations were depressed at initial insertion, followed by gradual increase to a baseline over 24 h. One possible explanation is that soil at the interface had been compacted by the corer, inhibiting gas exchange at the onset. Future applications of this system (e.g., calculating flux) will need to take this potential interference into account, likely via soil texture corrections [43].

Once the soil ecosystem re-stabilized, the CO_2_ level increased back to a normal state. It was also observed that the CO_2_ levels in the open land were higher than from under the tree. Furthermore, the CO_2_ levels were higher closer to the surface for open measurements and the opposite for under the tree. It is possible that tree root concentrations at 10 cm generated CO_2_, resulting in higher concentration relative to 10 cm probes in the open area. Furthermore, in probes located under the tree, there was evidence of a dip in CO_2_ concentration during the night across all days for probe D, and for probe C across days 3–5. This could reflect diurnal changes in tree root and root-associated microbial community respiration associated with diurnal change in carbon fixation and root exudation. Readings were also taken using the humidity (Figure 10) and temperature (Figure 11) sensors. The humidity sensors showed a slow increase in humidity levels before stabilizing around 90% RH. The temperature readings dropped rapidly down to ambient temperatures (around 12 °C), which was appropriate for autumn in the UK. The temperature sensors also measured the daily cycle in temperature (the probes were fitted at 11 am GMT time). It can be seen that the temperature cycles were more pronounced with the probes located closer to the surface over those embedded further into the ground.

The SGP40 VOC sensor was set to a low-power mode. This low-power mode limits the output to a raw sensor reading only, meaning the readings will not be adjusted to take into account the effect of temperature and humidity on the sensor. To make use of the readings, we pre-processed the raw VOC readings, which is the resistance of the sensing material of the sensor. The formula used for pre-processing is shown in Equation (1), where the ‘Average’ is the average of all the data collected. This allows easier comparisons of the data without the effect of the wide variation of raw sensor resistances.
Fractional Change = (Raw VOC − Average)/Average (1)

### 3.2. Deployment 2

The decrease in sample frequency from one per minute to one per two minutes for deployment 2 considerably improved the lifespan of the device, as we were able to gather up to 14 days of data compared with only 5 days in the first deployment. Additionally, Figure 13, Figure 14, Figure 15 and Figure 16 show that the switchover procedure resulted in a step change towards the partnering probe. This demonstrates the probe’s reliability and accuracy. Furthermore, we can see in all measurements that there is a diurnal pattern. The high temperature peak observed in Figure 15 was from the hottest day on record in the UK (at the time of this publication), where daytime temperatures reached close to 40 °C at the deployment.

### 3.3. Deployment 3

The results from the third deployment (Figure 17, Figure 18, Figure 19 and Figure 20) again show a diurnal pattern for all parameters. Perhaps most notable are the results shown in Figure 17, where on day 7 we can see a shift in flux between the 3 and 5 cm probes. As with the other deployments, apart from the open woodland set in the first deployment, there was evidence of higher CO_2_ levels deeper beneath the surface. For CO_2_ concentrations, humidity, and temperature, there were well-defined diurnal patterns, which were dampened at 15 cm relative to 3 and 5 cm. At 3 and 5 cm depth, peak CO_2_ concentrations were recorded at approximately 3 pm, while the lowest concentrations were measured at approximately 6 am. At 15 cm depths, peak CO_2_ concentrations were detected at midnight and the lowest concentrations were found at midday. The highest humidity was recorded at 6 am at 3 and 5 cm depth, and at midday at 15 cm depth, while the lowest humidity was measured at approximately 6 pm at 3 and 5 cm depths, and approximately 4 am at 15 cm depth. Diurnal rhythmicity of humidity persisted between days 1 and 8 but was not detected subsequently. Temperature patterns were conserved across 3, 5, and 15 cm depths, peaking at 4 pm and reaching their minimum between 2 and 4 am, with higher temperatures and greater diurnal range at 3 and 5 cm depth relative to 15 cm depth. VOC concentrations were not consistently different across soil depths, but there were diurnal differences in concentration across all depths, with the highest concentrations at 5–7 am and lowest concentrations at 2–3 pm.

The use of commercial CO_2_ probes in soils research is best highlighted by the National Ecological Observatory Network (NEON) throughout North America. In conjunction with other environmental measurements, Vaisala GMP-343 NDIR CO_2_ probes have been placed at various soil depths from 0 to 20 cm, replicated across 46 sites in continental USA, Hawaii, and Puerto Rico. To assess our probes against a commercial system, we compared publicly available soil CO_2_ data from three eastern USA NEON sites, with similar vegetation to the Wellesbourne woodland, against our concentrations from deployment three. NEON data was taken from the same time period (26 September–6 October 2022) [47]. We can see that when all CO_2_ concentration datapoints from within this timeframe are plotted as a function of soil depth, there is a clear trend of increasing CO_2_ concentration with depth (Figure 21). This is regardless of probe type, illustrating that our system compares well with commercial systems. Further, it could be argued that our system exceeds the capacity of the Vaisala GMP-343 by being completely wireless.

## 4. Discussion

In this paper, we describe the development of an IoT CO_2_ soil measurement system that is able to collect soil CO_2_ concentrations and other environmental parameters over an extended period and over a wide spatial area. Furthermore, our system is able to send back the data through a central gateway to an external site where the data can be downloaded and viewed, up to a distance of 300 m from the probes to the gateway. We have successfully demonstrated the functionality of our units within a woodland location in both open ground and under a tree. Our system here comprised of eight probes, but this could be increased significantly whilst still using a single gateway due to the low data rates of long-term monitoring. Current commercial systems cost in the order of USD 2 k–3 k per sensor module, thus an equivalent system to our array of eight units would cost in excess of USD 16 k. This does not include any costs associated with communication or the power system. The bill of materials for our system is currently around USD 250 per unit, but this does not take in account the cost of assembly and calibration, which is likely to add an extra USD 100 to the cost. To our knowledge, these are the first-battery powered WSN soil CO_2_ sensors currently reported. Our system also measures a wider range of environmental parameters (including VOCs) that are not available in existing commercial or previous research devices. Though water ingress in this set of experiments was not found to be a significant issue (beyond a single flooding event), further work will be needed to protect the sensors and the electronics from very harsh environments. In addition, the main LoRa gateway requires power to operate and therefore this component of the system is limited by this need. Our next version may have the gateway powered by a separate battery (or solar power) to produce a fully independent system, relying only on mobile phone signal to operate. Furthermore, our current unit is limited to only 14–20 days of use. For longer running, it may be necessary to replace the microcontroller with a lower-power version, reduce SD card writing and add an option for solar charging. With such changes, continual operation across several months should be achievable and will be limited by the drift in the CO_2_ sensor. Table 3 gives a comparison of our unit compared to the current commercial systems on the market.

Our system has the capacity to detect spatial and temporal changes in CO_2_ and VOC concentrations, humidity, and temperature, and these were clearest in deployment three. CO_2_ concentrations in soil reflect plant root respiration and the respiration of heterotrophic soil biota utilising plant root exudates and organic matter as substrates. Plant root respiration is known to exhibit diurnal rhythmicity associated with diurnal cycles of carbon fixation and the subsequent translocation of assimilate to roots [48]. Similarly, exudation of soluble sugars and organic acids by roots may also be diurnally rhythmic, driving diurnal changes in microbial community composition and activity in the root zone [49]. Additionally, temperature will exert control over both root and microbial metabolism. The systems therefore had the capacity to detect temporal and vertical CO_2_ concentration differences arising from root distribution profiles, microbial activity, and temperature gradients. In these experiments the probes were deployed at depths less than 20 cm to align with the NSF Neon Soil CO_2_ measurement specifications. However, the probes could equally be used at much greater depths which would allow their use in monitoring other ecosystems, with minimal modification (for example, extending the aerial and adding a method of recovering the probe). Finally, the focus here was on the development of the unit and to show its functionality across a common habitat in the UK: woodland. In our next steps, we will first make in situ comparisons between our probes and a validated CO_2_ monitoring system, exploring both passive concentration comparisons and our ability to estimate CO_2_ flux from the soil to the atmosphere. Similarly, we will investigate the long-term performance and resilience of the probes for both functionality and accuracy under various experimental conditions. Finally, we will undertake more experiments in an agricultural context, where we focus on the data generated by the system and align it with soil health.

## 5. Conclusions

Here, we have successfully demonstrated the design, development, and deployment of a CO_2_ IoT system for monitoring soil health. We have produced eight sensor probes that were able to measure soil CO_2_ for a period up to 14 days and send the data back through a LoRa gateway/mobile phone network to an external site. The units also successfully measured soil temperature and humidity and were able to measure the daily changes in temperature close to the surface. The probes were placed at a range of depths underneath the surface and were able to measure distinct diurnal differences in CO_2_ levels. In open ground, the CO_2_ was higher towards the surface, whilst under trees, the CO_2_ was higher deeper into the ground. Our next steps are to use these probes in an agricultural setting to understand more about soil health in these environments.

## Figures and Tables

**Figure 1 sensors-23-02580-f001:**
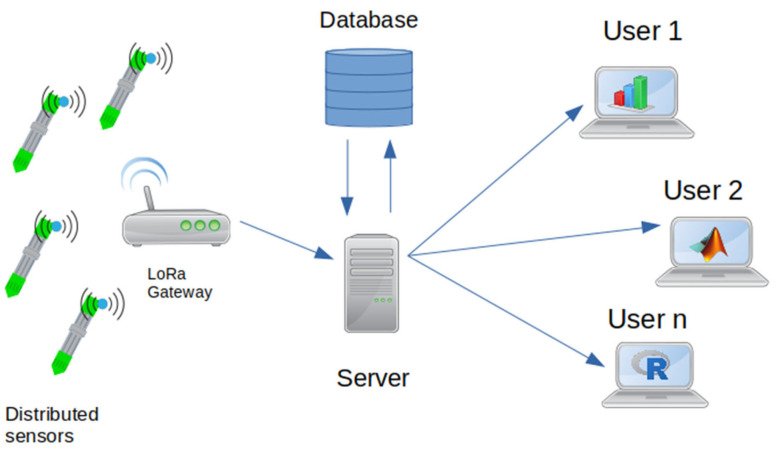
Distributed network concept.

**Figure 2 sensors-23-02580-f002:**
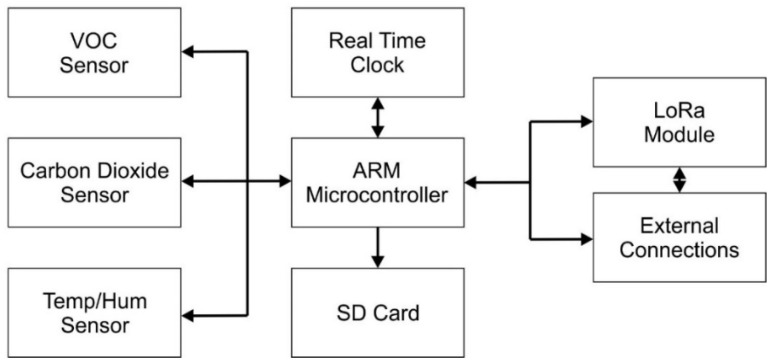
System block diagram.

**Figure 3 sensors-23-02580-f003:**
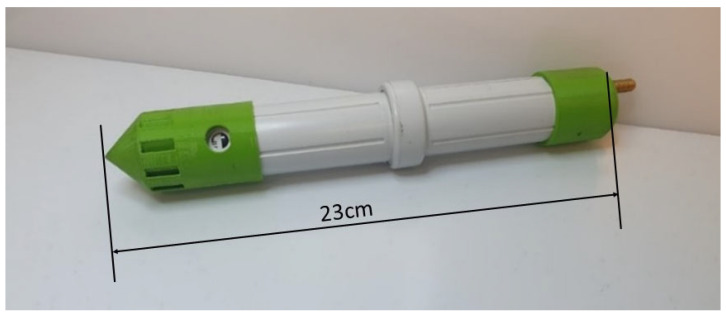
Probe casing with end caps designed for soil submersion.

**Figure 4 sensors-23-02580-f004:**
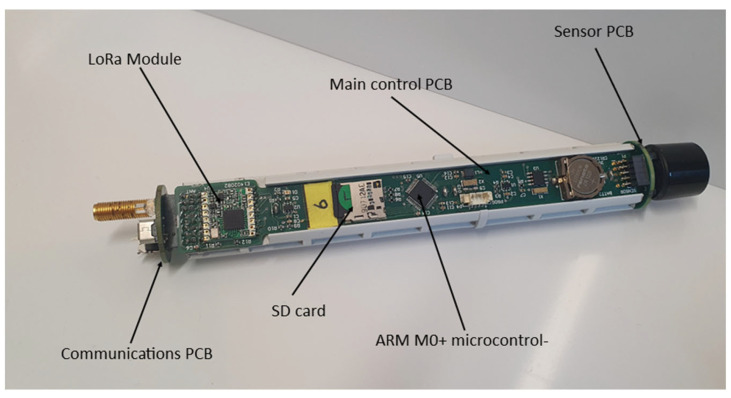
Internals configuration of the sensor probe.

**Figure 5 sensors-23-02580-f005:**
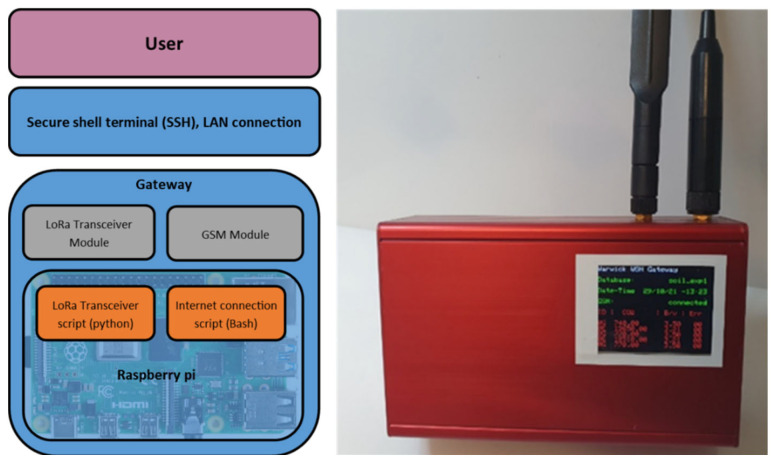
Gateway block diagram and physical setup.

**Figure 6 sensors-23-02580-f006:**
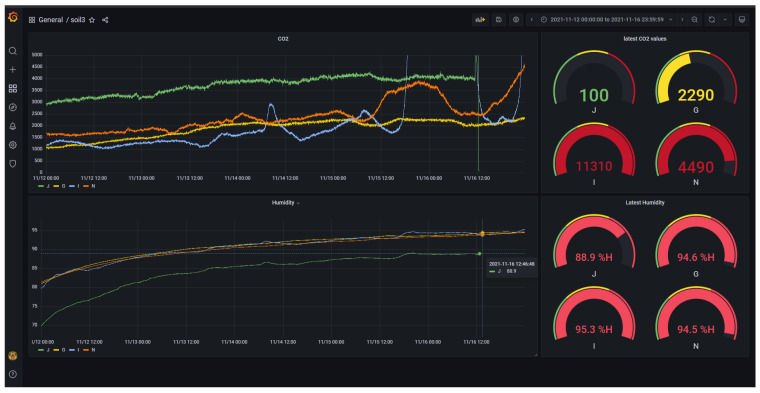
Live data dashboard viewed through Chrome internet browser.

**Figure 7 sensors-23-02580-f007:**
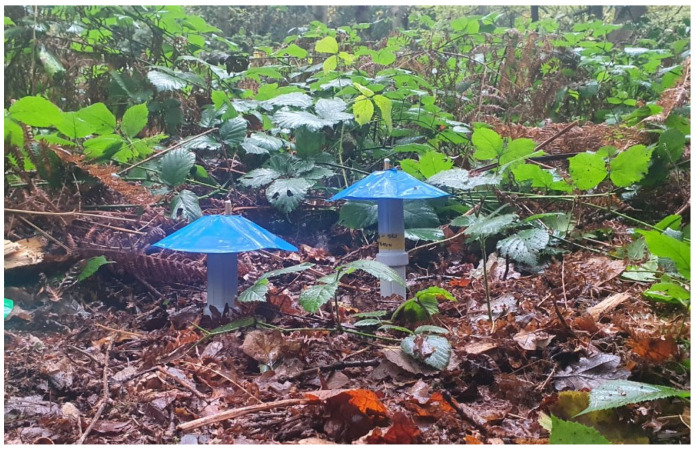
CO_2_ probes driven to different soil depths.

**Figure 8 sensors-23-02580-f008:**
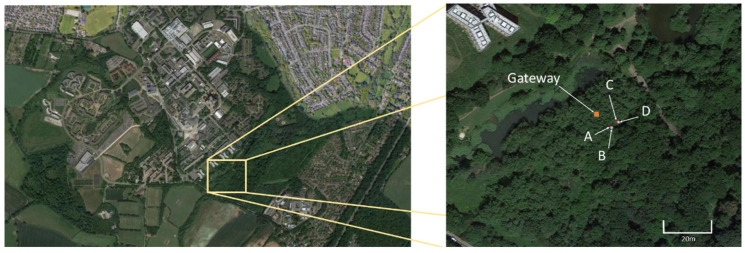
Satellite image of device location [Google Maps, accessed 29 March 2022]. Probes A&B in open woodland, C&D placed under tree.

**Figure 9 sensors-23-02580-f009:**
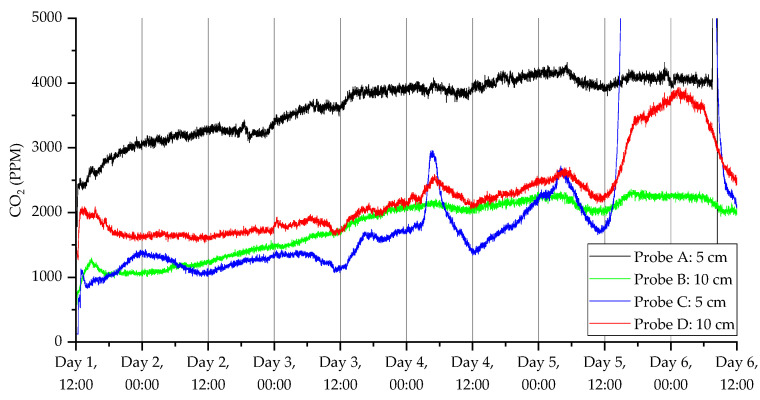
CO_2_ measurements for the deployment in Warwick campus woodland. Data collection commenced on 11 November 2021 at 12:40 (24 h time).

**Figure 10 sensors-23-02580-f010:**
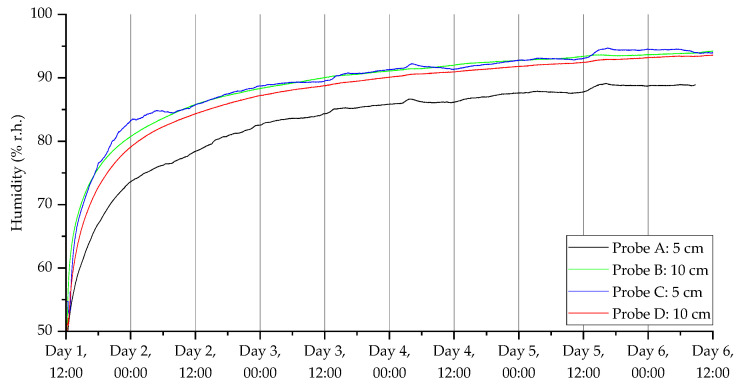
Humidity measurements for the deployment in Warwick campus woodland. Data collection commenced on 11 November 2021 at 12:40 (24 h time).

**Figure 11 sensors-23-02580-f011:**
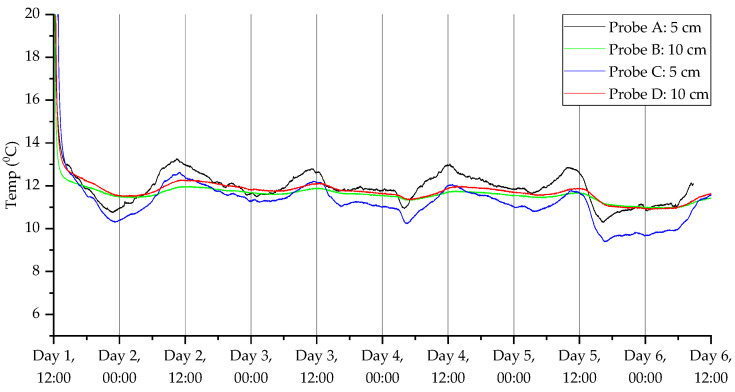
Temperature measurements for the deployment in Warwick campus woodland. Data collection commenced on 11 November 2021 at 12:40 (24 h time).

**Figure 12 sensors-23-02580-f012:**
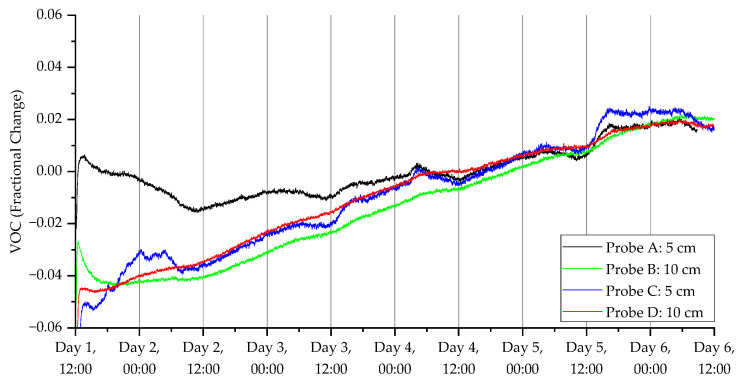
VOC measurements for the deployment in Warwick campus woodland. Data collection commenced on 11 November 2021 at 12:40 (24 h time).

**Figure 13 sensors-23-02580-f013:**
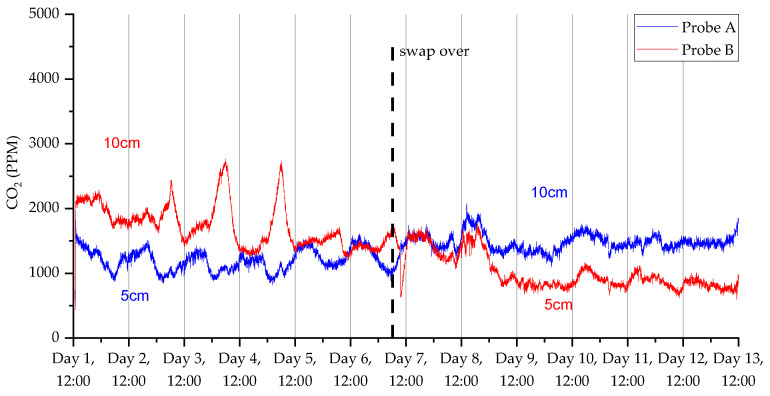
CO_2_ measurements for the 2nd deployment in Wellesbourne campus woodland. Data collection commenced on 12 February 2022 at 12:45 (24 h time).

**Figure 14 sensors-23-02580-f014:**
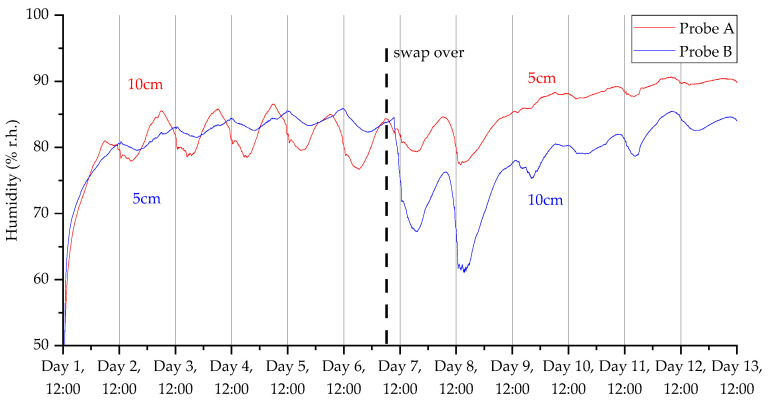
Humidity measurements for the 2nd deployment in Wellesbourne campus woodland. Data collection commenced on 12 July 2022 at 12:45 (24 h time).

**Figure 15 sensors-23-02580-f015:**
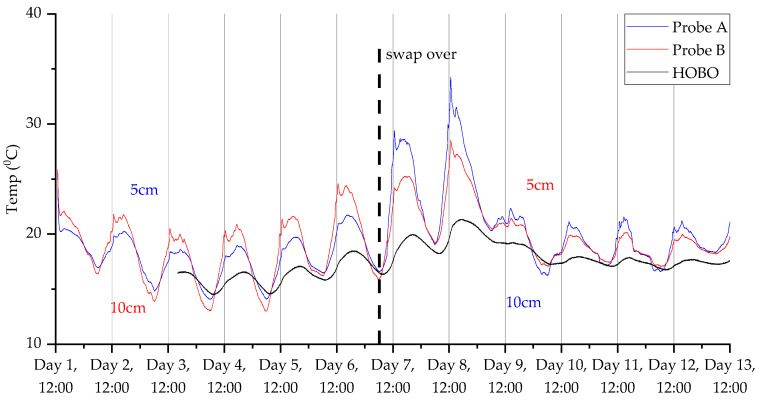
Temperature measurements for the second deployment in Wellesbourne campus woodland. Data collection commenced on 12 July 2022 at 12:45 (24 h time).

**Figure 16 sensors-23-02580-f016:**
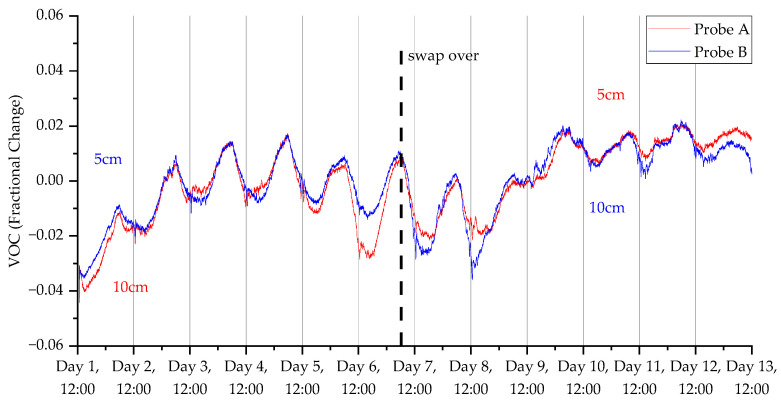
VOC measurements for the second deployment in Wellesbourne campus woodland. Data collection commenced on 12 July 2022 at 12:45 (24 h time).

**Figure 17 sensors-23-02580-f017:**
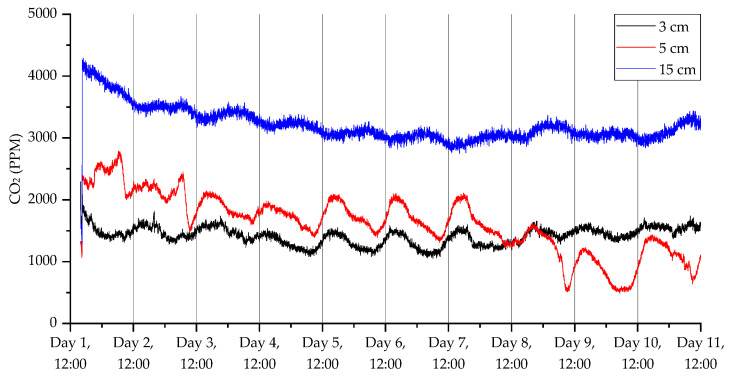
CO_2_ measurements for the third deployment in Wellesbourne campus woodland. Data collection commenced on 26 August 2022 at 16:00 (24 h time).

**Figure 18 sensors-23-02580-f018:**
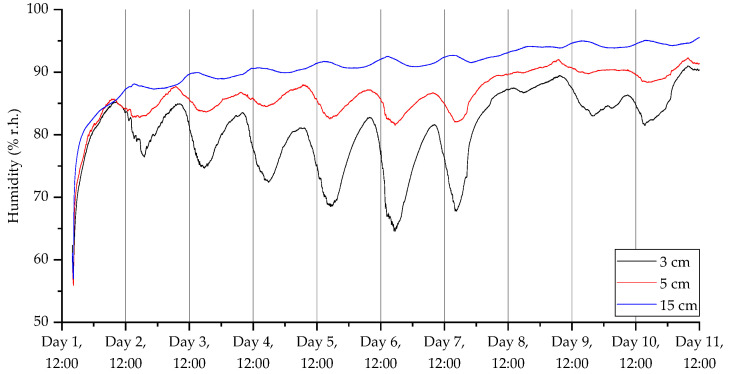
Humidity measurements for the third deployment in Wellesbourne campus woodland. Data collection commenced on 26 August 2022 at 16:00 (24 h time).

**Figure 19 sensors-23-02580-f019:**
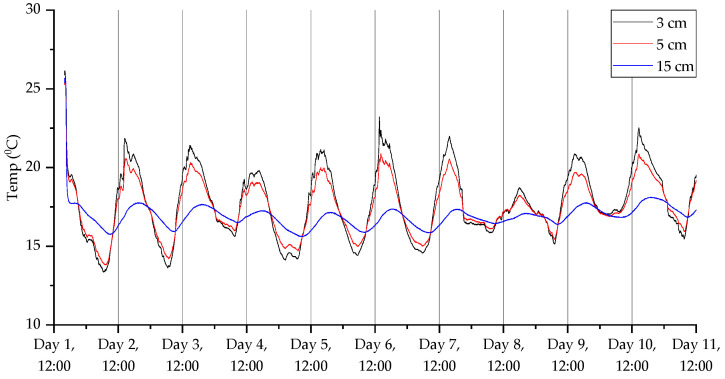
Temperature measurements for the third deployment in Wellesbourne campus woodland. Data collection commenced on 26 August 2022 at 16:00 (24 h time).

**Figure 20 sensors-23-02580-f020:**
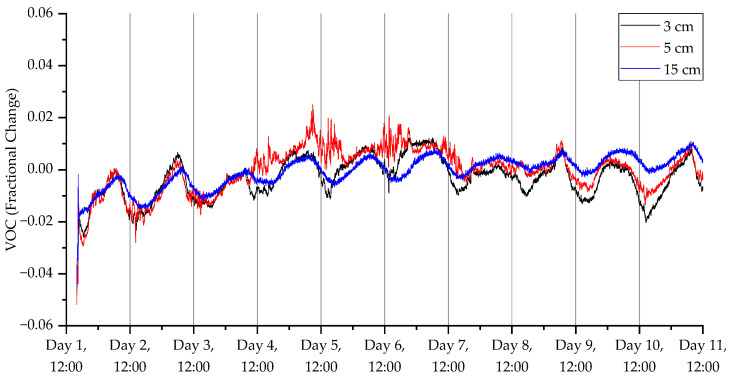
VOC measurements for the third deployment in Wellesbourne campus woodland. Data collection commenced on 26 August 2022 at 16:00 (24 h time).

**Figure 21 sensors-23-02580-f021:**
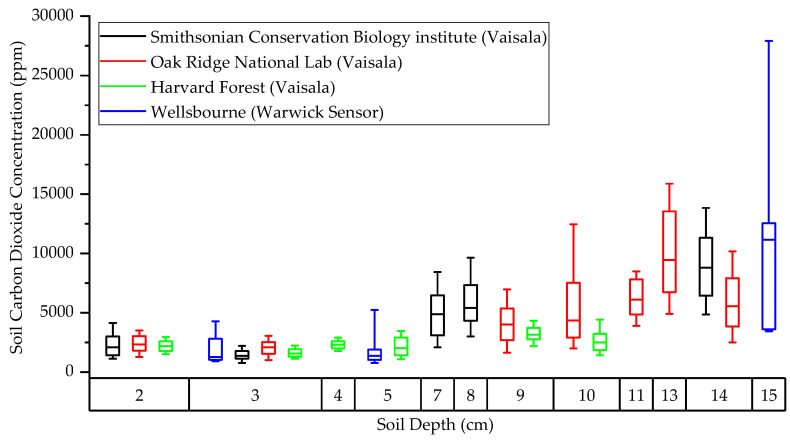
Box plot data comparisons between research institutes at similar times of year.

**Table 1 sensors-23-02580-t001:** Main components used in the sensor probe.

Part/Supplier	Specification	Purpose
ExplorIR-M, GSS, UK	0–5% CO_2_ ±(70 ppm, +5%)	Carbon dioxide NDIR Sensor
SHT31, Sensirion, Switzerland	±2 @0–100% RH ±0.2 @0–90 °C	Temperature and humidity sensor
SGP40 Sensirion, Switzerland	0 to 1000 ppm of ethanol equivalents	Total VOC sensor
ATSAMD21G, Microchip	ARM M0+	Microcontroller
PCF8523T/NXP	Year, month, day, weekday, hours, minutes, and seconds	Real-time clock
RFM96W, Sparkfun	LoRa 433 MHz	Communication unit

**Table 2 sensors-23-02580-t002:** Table summary of programs.

Device	Program	Function	Access
Probe	Data logger	Log data from sensor Send using Lora comms	Local, USB—serial port
Gateway	LoRa transceiver	To receive data from probes To separate data into headings and data types Send data using http request to API	One-way, updated by downloading remote repository Secure shell terminal, LAN connection
Gateway	Internet connection	Monitor internet connection Renew IP lease upon expiry	One-way, updated by downloading remote repository Secure shell terminal, LAN connection
Server	API	Check legitimacy of incoming data Send conformation back to gateway Insert data to correct table Manage data requests from website	Secure shell login, wan
Server	MySQL Database	Store data	Terminal login DBeaver user interface MySQL workbench
Server	Website	View data from database	Secure shell login ftp server Fasthosts dashboard

**Table 3 sensors-23-02580-t003:** Comparison of our reported soil probe system to commercial systems.

	eosGP CO_2_ [48]	Vaisala GMP343 [47]	Warwick CO_2_ Probe
Dimensions [Len, Dia]	216 mm × 51 mm	194 mm × 55 mm	230 mm × 32 mm
Weight	400 g	360 g	172 g
Operating power	400 mW	<3500 mW	~20 mW
Sensor technology	NDIR	NDIR	NDIR
Sensor Range	0–30,000 ppm	0–20,000 ppm	0–50,000 ppm
Accuracy	±3.5% of reading	±5 ppm ±2% of reading	±70 ppm
Communication Interface	RS485/analogue 0–5 v	RS485/RS232	LoRa/UART USB
Battery life	Mains	Mains	14 Days
Response time T90	<90 s	82 s	22 s

## Data Availability

The raw data is available at: http://wrap.warwick.ac.uk/view/author_id/9191.html, accessed on 7 December 2022.

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
