# Peer review of "Integrated Probe System for Measuring Soil Carbon Dioxide Concentrations"

_sensors, 2023, doi:10.3390/s23052580_

Round 1
Reviewer 1 Report
This paper developed a CO2 IoT system for soil CO2. The results have demonstrated the concentration of CO2 change for a period of 14 days.
1. The results have to be compared to other CO2 measurements which were reported previously, the advantages of your system should be compared with other studies.
2. How does the CO2 change affect agriculture and palnts, what is the relationship between them?
Reviewer 2 Report
General comments:
The present manuscript is a solid work describing the development and tests of an in-situ soil carbon dioxide probe system. The paper is well structured and the reader gets insights in the design and the potential application fields for such a monitoring system.
Soil gas monitoring has a tremendous range of applications not only for biogeochemical research questions (as mentioned and well described in the introduction paragraph). The determination of CO2 in soils and sediments plays also an important role in volcanology, fault detection and furthermore for technical monitoring in the field of Carbon Capture and Sequestration (CCS) in terms of safety reasons due to potential leakages. Providing robust and reliable monitoring systems based on low-cost approaches would close a huge existing gap. The here presented monitoring system may have that capacity to detect spatial and temporal changes in CO2 and soil physical properties such as humidity and temperature.
However, the deployment examples shown are not as clear as to be convincing. Therefore, I recommend a major revision of the current version.
More detailed recommendations:
Section 2 – Materials and Methods
Measurement depths of 15 cm are very shallow. Especially depths lesser than 5 cm are too shallow for a reliable soil gas monitoring due to the non-negligible influence of the ambient atmosphere at ground surface (and the possibility to create short circuits with the surface along the probe housing). How does your system avoid the mixing of soil gas and atmospheric influence?
An extension to greater depths (larger than 30 cm) would be of enormous importance for an optimal and widely applicable system. Soil gas monitoring (for biogeochemical processes) is often coupled with soil moisture measurements, which mainly cover the soil horizons down to 50 – 80 cm depth (zone where you can find roots and respiration). Another important issue I would recommend is to add to the system a sensor to observe the atmospheric pressure, which is an extremely important factor controlling the land-atmosphere exchange and the emission behaviour of the soil. Also soil moisture is of great importance for soil gas emission (see above) and it could be helpful to couple it with a soil moisture sensor.
Figure 3 -5 show the probe and the gateway. However, I missed the information on the dimension of the probe (diameter and weight) and of the gateway, which could be easily added to the figures.
For experimental reasons the authors mentioned a sampling frequency of 2 minutes which led to an observing period of 14 days (without battery maintenance). Maybe here I see potential for further extension of observation times. Due to the low frequency of soil physical / chemical processes a sampling frequency of 5 minutes might be perfectly adequate.
Section 2.4 / 3 – Deployment and Results
Here I have two main concerns.
1. Please specify the soil type and the vegetation of your test sites. Detailed information on the soil physical properties (sand, silt, clay, …) and the kind of vegetation as well as root density are driving factors for soil gas concentration. It is crucial necessary to evaluate your data examples.
2. A comparison of your system with other measurement options (e.g. commercial system or gas samples and laboratory analysis) would be helpful to evaluate your system also in terms of data quality and trustworthiness. The simple comparison of data from sites in USA (”similar” to your site) is less convincing. What means similar? How about the environmental conditions? Figure 21 shows the problems with soil gas measurements in depths lesser than 5 cm and the known relation of increasing CO2 concentrations with depth. That is not a proof of concept of your devices. It would also be helpful to have longer time series at the same place under changing environmental conditions – for instance to show that there are no drift effects.
I see the potential of the presented system, but for a persuasive representation the manuscript still needs some meaningful data.
Round 2
Reviewer 1 Report
No further questions.
Author Response
We would like to thank reviewer 1 for their input.
Reviewer 2 Report
Dear authors,
I still have my concerns concerning the data quality of your system, which should be checked e.g. in direct comparison to other measurement options or systems (which is a typical engineering approach). I see your point in terms of time and effort to realize further measurements. However, I expect some words at the end of your discussion section taken this open issue into account - not only the phrase "the focus here was on the development of the unit and to show its functionality across habitats" (what is not right - it is all the same habitat type forest) – we will now look to validate the system in an agricultural applied context." Here the reader would expect some words concerning the next steps on long-term validation and comparison to other comercial systems.
